

# Virulence characteristics of *Blumeria graminis* f. sp. *tritici* and its genetic diversity by EST-SSR analyses

Yazhao Zhang[1],[*], Xianxin Wu[1],[*], Wanlin Wang[1], Yiwei Xu[1], Huiyan Sun[1], Yuanyin Cao[1], Tianya Li[1] and Mansoor Karimi-Jashni[2]

[1] College of Plant Protection, Shenyang Agricultural University, Shenyang, China
[2] Department of Plant Pathology, Tarbiat Modares University, Tehran, Iran
[*] These authors contributed equally to this work.

Corresponding authors
Tianya Li, litianya11@syau.edu.cn
Mansoor Karimi-Jashni,
mkjashni@gmail.com

## ABSTRACT

Wheat powdery mildew, caused by *Blumeria graminis* f. sp. *tritici* (an obligate biotrophic pathogen) is a worldwide threat to wheat production that occurs over a wide geographic area in China. For monitoring genetic variation and virulence structure of *Blumeria graminis* f. sp. *tritici* in Liaoning, Heilongjiang, and Sichuan in 2015, 31 wheat lines with known Powdery mildew resistance genes and 2 EST-SSR markers were used to characterize the virulence and genetic diversity. Results indicated that 90% of all isolates were virulent on *Pm3c*, *Pm3e*, *Pm3f*, *Pm4a*, *Pm5*, *Pm6* (Timgalen), *Pm7*, *Pm16*, *Pm19*, and *Pm1 + 2 + 9* and 62.6% to 89.9% of isolates were virulent on *Pm3a*, *Pm3b*, *Pm3d*, *Pm4b*, *Pm6* (Coker747), *Pm8*, *Pm17*, *Pm20*, *Pm23*, *Pm30*, *Pm4 + 8*, *Pm5 + 6*, *Pm4b + mli*, *Pm2 + mld*, *Pm4 + 2X*, *Pm2 + 6*. The *Pm13* and *PmXBD* genes were effective against most collected isolates from Liaoning and Heilongjiang Provinces. Only *Pm21* exhibited an immune infection response to all isolates. Furthermore, closely related isolates within each region were distinguished by cluster analyses using EST-SSR representing some gene exchanges and genetic relationships between the flora in Northeast China (Liaoning, Heilongjiang) and Sichuan. Only 45% of the isolates tested show a clear correlation between EST-SSR genetic polymorphisms and the frequency of virulence gene data. However, the EST-SSR polymorphism of isolated genes did not correspond to the virulence diversity of isolates in the single-gene lineage identification of hosts.

## INTRODUCTION

Wheat powdery mildew, caused by *Blumeria graminis* f. sp. *tritici* (*Bgt*), is one of the major diseases affecting the production of wheat in China. *Bgt* is widely distributed and displays a complex population structure and rapid mutation rate, which make the disease extremely virulent and difficult to prevent and control (*Abdelrhim et al., 2018*; *Cowger et al., 2018*). Using cultivars carrying major resistance genes (known as *Pm* genes) is the most economical and cost-effective way to control wheat powdery mildew (*Lu et al., 2020*; *Parks et al., 2008*). To date, more than 91 *Pm* resistance genes, mapped to 58 loci, have been identified that confer resistance to wheat powdery mildew (*Nanjundan et al.,*

2020; *Petersen et al., 2015*; *Tan et al., 2019*; *Xu et al., 2020*). However, resistant cultivars carrying *Pm* genes confer complete resistance to specific *Bgt* races leading to high pressure on fungus. Due to genetic variations occurring in fungal races, wheat cultivars carrying effective resistance genes are not able to recognize *Bgt* races anymore and turn into susceptible cultivars (*Wolfe & Schwarzbach, 1978*; *Wu et al., 2019*). Monitoring of population dynamic and genetic variation analysis of *Bgt* provide a basis for timely warning and sustainable management of wheat powdery mildew using disease-resistant cultivars.

In China, wheat powdery mildew disease mainly occurs in the Yunnan-Guizhou-Sichuan zone, the wheat region in the middle and lower reaches of the Yangtze River, and the wheat region in Huang-Huai-Hai Region and Ningxia, Inner Mongolia, and Northeast China (*Sheng et al., 1995*). The spread of wheat powdery mildew in China also occurs from south to north over long distances in spring, and in the opposite direction in autumn (*Wu et al., 2019*). Northeast China has cold winters that last more than 6 months, therefore the local wheat powdery mildew *Bgt* does not survive as primary inoculum for the following year. The initial inoculum for the spring season of Northeast China originates from southern China, the Jiaodong Peninsula region (*Yang et al., 1992*). The data of inter simple sequence repeat (ISSR) analysis confirmed that Shandong, Henan, Hubei, and especially Shandong provided the initial inoculum of *Bgt* for the spring wheat area of Northeast China (*Zhu et al., 2015*).

Assessment of infection type on wheat differential lines is the most basic method to identify the races of *Bgt* and to analyze virulence genes virulence evolution of pathogenic populations (*Imani, Ouassou & Griffey, 2002*; *Liu et al., 2015*; *Niewoehner & Leath, 1998*). However, with the development of molecular biology techniques, DNA molecular markers have been used extensively in the analysis of *Bgt* population evolution (*Comlekcioglu et al., 2010*; *Shao, Xu & Chen, 2011*; *Zhu et al., 2010*). Our previous analysis revealed that isolates collected in 2013 and 2014 from Northeast and Northwest China (Gansu, Heilongjiang, and Liaoning), have a clear genetic relationship (*Wu et al., 2019*). In this study, the genetic diversity of these *Bgt* isolates was analyzed using Expressed sequence tag-simple sequence repeat (EST-SSR) molecular markers to explore the genetic structure of the *Bgt* population and the relationship between different populations in the two regions. The data obtained from the genetic structure and the distribution of *Bgt* isolates will be discussed.

## MATERIALS AND METHODS

### Collection of *Bgt* isolates

From May to July 2015, diseased leaves with fresh powdery mildew spores were collected from Liaoning and Heilongjiang and kept in falcon tubes containing water agar medium with 1% of 6-benzylaminopurine preservation solution (40 mg·L$^{-1}$). Diseased leaves carrying the cleistothecium of *Bgt* were also collected from Sichuan, brought back to the laboratory, dried in a cool place, and stored at 4 °C. Xianxin Wu and Wanlin Wang identified all isolates used in the study. All isolates were deposited (no deposition number) in the College of Plant Protection, Shenyang Agricultural University (our lab).

## Isolation, purification, and propagation of *Bgt*

The highly susceptible Little Club (provided by the College of Plant Protection of Shenyang Agricultural University) was sowed in tile pots. When the seedlings had grown to the one-leaf stage, 5–6 cm leaf segments were cut and placed in a Petri dish lined with double filter paper. Five to six-leaf segments were placed face up in each Petri dish and held in place with glass strips on both ends. The filter paper was moistened with 40 mg·L$^{-1}$ of 6-benzylaminopurine (6-BA) preservation solution. The collected powdery mildew was first attached to the front of the leaf with a toothpick sharpened into a flattened tip. Three spots were attached to each leaf, lightly and evenly applied taking care not to scratch the leaf. After inoculation, the leaves were incubated for 5–7 days in a growth chamber at 18–22 °C with a 14-h/10-h light/dark cycle. After white spore mounds appeared on the leaves, the leaves were inoculated by shaking. The spores on one leaf segment were gently shaken off on the freshly isolated leaf segment. After incubation at 18–22 °C for 5–7 days under light, single colonies were inoculated at three points on the isolated leaf segments with a flat toothpick, and this process was repeated several times for the isolation and purification of monospores. The isolated and purified single pustules were numbered and multiplied for preservation (*Zhu et al., 2015*).

## Release of *Bgt* cleistothecium ascospores

A cleistothecium was picked from diseased leaves using an inoculating needle and incubated on moistened filter paper soaked with distilled water. After 5 days, the cleistothecium was picked and observed under a microscope. When ascospores were formed, a cleistothecium was randomly picked from diseased leaves and transferred to moistened filter paper sheets, with one cleistothecium per sheet. The filter paper sheet was then placed upside down in the center of a Petri dish lid, the Petri dish lid was placed on the petri dish with the isolated wheat leaves, and the Petri dish was sealed with parafilm and placed in a light incubator at 17 °C (*Chi, 2009*).

## Analysis of virulence frequency of *Bgt*

A total of 31 lines with known *Pm* genes (provided by the College of Plant Protection of Shenyang Agricultural University) were sown in 10 cm diameter tile pots according to their numbers and marked. The highly susceptible variety Chancellor was used as the susceptible control. The first leaf was cut when the seedlings reached the one-leaf stage and the cut leaf segments were placed in order in the petri dishes, and the propagated single pustules were gently shaken off the leaves. The culture was incubated for 5–7 days. When the susceptible control was fully developed, the infection types (ITs) were assessed and recorded following the method described by the previous study (*Si et al., 1987*): ITs 0–2 were marked as resistant (R), and the corresponding isolates as avirulent; ITs 3–4 were marked as susceptible (S) and the corresponding isolates as virulent. The test was repeated three times.

**Table 1  Seven EST-SSR primer pairs suitable for analyzing the DNA polymorphism in *B. graminis* f. sp. *tritici*.**

| Codes | Primers | Sequences | Repeat motif | Annealing temperature (°C) | Polymorphism |
|---|---|---|---|---|---|
| 1-F | Blu SSR3-1 | TTCGAGGCAAGCTCTTCTCA | $(CCGTTC)_4$ | 56 | Polymorphic |
| 1-R | Blu SSR3-2 | TTTCGGCAGGCAAGTTTATT | | | |
| 2-F | Blu SSR18-1 | GGGTAACGATTGGTTAGGTGCT | $(ATCACC)_3$ | 56 | Monomorphic |
| 2-R | Blu SSR18-2 | AGGTGGTGGTAAAGGGGATGAT | | | |
| 3-F | Blu SSR29-1 | GGAGGATCGGTAGCAGTG | $(GCA)_5$ | 56 | Monomorphic |
| 3-R | Blu SSR29-2 | GCGGCGGTAGCTTCTTTT | | | |
| 4-F | Blu SSR32-1 | GGGGAGGTATAGGTGAGG | $(TCT)_6$ | 52 | Polymorphic |
| 4-R | Blu SSR32-2 | GAGCGTTTGCTGTTCTGT | | | |
| 5-F | Blu SSR35-1 | AGACTCACAGCAGAGCAAA | $(CTTCAA)_3$ | 52 | Monomorphic |
| 5-R | Blu SSR35-2 | GCAGATCCATGATCTTCGT | | | |
| 6-F | Blu SSR41-1 | ATCCATTGTAGTTAGGAGCCA | $(AC)_6$ | 54 | Monomorphic |
| 6-R | Blu SSR41-2 | ATGACCTGATTGATTTATCCC | | | |
| 7-F | Blu SSR44-1 | TGAGGATTTAGATGATATGGA | $(AGA)_5$ | 52 | Monomorphic |
| 7-R | Blu SSR44-2 | GATCTTAAATTATTTTGACCG | | | |

## Genomic DNA extraction and polymerase chain reaction (PCR) analysis

Genomic DNA was extracted from conidia using the Omega Bio-Tek fungal DNA kit (Norcross, GA, USA) following the manufacturer's protocol. The total PCR reaction volume was 20 μL, consisting of 1 μL DNA template (30 ng·μL$^{-1}$), 1 μL forward primer, 1 μL reverse primer, 10 μL 2 × Power *Taq* PCR Master Mix, and 7 μL ddH$_2$O. The PCR procedure was as follows: initial denaturation at 94 °C for 5 min; then 35 cycles of a denaturation step at 94 °C for 30 s and an extension step at 72 °C for 1.5 min, followed by a final extension at 72 °C for 10 min.

## Selection of the EST-SSR primers

Seven pairs of EST-SSR primers (Table 1) were designed according to *Xu (2012)* and were screened for the occurrence of clear and stable polymorphisms. Two pairs were chosen for the genetic polymorphism analysis. The primers were synthesized by Sangon Biotech Inc. (Shanghai, China; http://www.sangon.com/).

## Polyacrylamide gel electrophoresis and genetic diversity analysis

The procedure for polyacrylamide gel electrophoresis (PAGE) was as previously published (*Chen et al., 2015*). The silver staining method was used to visualize the PCR products as described by *Bassam, Caetano-Anollés & Gresshoff (1991)*. Based on the PAGE results, '1' or '0' were assigned to the presence or absence of bands, respectively, and the same method was used for frequency of virulence analysis, '1' or '0' were assigned to resistance (ITs: 0–2) or susceptibility (ITs: 3–4), respectively, in the host. According to the '1, 0' data matrix, the genetic similarity was calculated using NTSYSpc 2.10$_e$. The unweighted pair group arithmetic method was used for gene diversity cluster analysis of the expression sequences and then the classification trees were constructed. In addition, the genetic and virulence correlations of different isolates were described by Principal Component

Analysis (PCA) analysis. The isolates were divided into three groups by 'K-mean' method. The optimal number of groups was determined by the 'gap_stat' method in 'cluster' package of R software 4.0.0.

## RESULTS

### Virulence frequencies of 80 isolates to 36 single gene lines

A total of 80 *Bgt* isolates, collected from Liaoning (29), Heilongjiang (26), and Sichuan (25), were isolated (Table S1). Virulence frequencies of these isolates were assessed on 36 differential lines individually containing single powdery mildew (*Pm*) resistance gene. The results show that the virulence frequency of isolates from Northeast China against resistance genes including *Pm1*, *Pm3c*, *Pm3d*, *Pm3e*, *Pm3f*, *Pm4a*, *Pm4b*, *Pm5*, *Pm6* (Coker747), *Pm6* (Timgalen), *Pm7*, *Pm8*, *Pm16*, *Pm17*, *Pm19*, *Pm23*, *Pm30*, *Pm4 + 8*, *Pm4b + mli*, *Pm4 + 2X*, and *Era* was above 60%, indicating that the effectiveness of these resistance genes had been partially or completely lost. Virulence to *Pm 3a*, *Pm3b*, *Pm13*, *Pm18* (*1c*), *Pm20*, *Pm22* (*1e*), *Pm5 + 6*, and *PmV2 + 6* was 40–60%. Virulence to *PmVXBD* and *Pm21* were 27.5% and 0 (Table 2).

### EST-SSR analysis of *Bgt* isolates

Two pairs of EST-SSRs, Blu ssr3-1-blu ssr3-2 and Blu ssr2-1-blu ssr32-2, were selected from seven reported EST-SSR primers, 9 and 20 bands of clear and stable polymorphisms were amplified respectively. These two pairs of primers were specific to *Bgt* isolated from Northeastern China and Gansu Province. Figures 1 and 2 show the PAGE results of EST-SSR polymorphism using primer pairs Blu SSR3-1- plus Blu SSR3-2 and Blu SSR32-1- plus Blu SSR32-2, respectively. Genetic similarity analysis of EST-SSR PAGE was assessed on 80 *Bgt* isolates collected from Liaoning, Heilongjiang, and Sichuan using NTSYSpc 2.10$_e$ and PCA. NTSYSpc 2.10$_e$ dates analysis showed when the genetic similarity coefficient was 0.52, the 80 isolates were divided into three groups: Group I consisted of 36 isolates, including 23 isolates from Sichuan, nine isolates from Liaoning, and four isolates from Heilongjiang (Fig. 3); Group II consisted of 38 isolates, including 20 isolates from Heilongjiang, 16 isolates from Liaoning, and two isolates from Sichuan; Group III consisted of six isolates, including four isolates from Liaoning, and two isolates from Heilongjiang. PCA dates analysis is similar to NTSYSpc 2.10$_e$, and all isolates were also divided into three groups (Fig. 4). The PCA model described 26.5% variance. 15.1% variance was described by Dim 1, and 11.4% was by Dim 2. Overall, there was a certain degree of transmission among *Bgt* isolates in different regions. However, when the genetic similarity coefficient was high, some isolates from Heilongjiang and Liaoning were clustered into subcategories. In contrast, isolates from Sichuan were clustered into subcategories separately, indicating that the genetic exchange between isolates from Heilongjiang and Liaoning was extensive. At the same time, it showed that there were genetic differences among these *Bgt* groups from different regions.

**Table 2 Occurrence frequency of virulence genes in *B. graminis* f. sp. *tritici* in 2015.**

| Cultivar (line) | *Pm* gene | Virulence gene | Occurrence frequency of virulence genes (%) | | | |
|---|---|---|---|---|---|---|
| | | | Heilongjiang | Liaoning | Sichuan | Average |
| Axminster/8cc | *Pm1* | *V1* | 74.1 | 93.1 | 95.8 | 87.7 |
| Asosan/8cc | *Pm3a* | *V3a* | 55.5 | 58.6 | 75.0 | 63.0 |
| Chul/8cc | *Pm3b* | *V3b* | 48.1 | 51.7 | 100.0 | 66.6 |
| Sonora/8cc | *Pm3c* | *V3c* | 100.0 | 96.6 | 100.0 | 98.9 |
| Kolibri | *Pm3d* | *V3d* | 88.9 | 65.5 | 50.0 | 68.1 |
| W150 | *Pm3e* | *V3e* | 96.3 | 96.6 | 100.0 | 97.6 |
| Michi. Amber/8cc | *Pm3f* | *V3f* | 100.0 | 93.1 | 95.8 | 96.3 |
| Khapli/8cc | *Pm4a* | *V4a* | 96.3 | 92.0 | 95.8 | 94.7 |
| Armada | *Pm4b* | *V4b* | 63.0 | 58.6 | 91.7 | 71.1 |
| Hope/8cc | *Pm5* | *V5* | 92.6 | 93.1 | 91.7 | 92.5 |
| Timgalen | *Pm6* | *V6* | 100.0 | 95.6 | 87.5 | 94.4 |
| Coker747 | *Pm6* | *V6* | 74.1 | 79.3 | 100.0 | 84.5 |
| CI14189 | *Pm7* | *V7* | 100.0 | 96.6 | 100.0 | 98.9 |
| Kavkaz | *Pm8* | *V8* | 33.3 | 86.2 | 95.8 | 71.8 |
| R4A | *Pm13* | *V13* | 18.5 | 22.1 | 28.3 | 23.0 |
| Brigand | *Pm16* | *V16* | 92.6 | 89.7 | 91.7 | 91.3 |
| Amigo | *Pm17* | *V17* | 88.9 | 72.4 | 95.8 | 85.7 |
| MIN | *Pm18 (Pm1c)* | *V18 (V1c)* | 54.5 | 41.4 | 54.2 | 50.0 |
| XX186 | *Pm19* | *V19* | 100.0 | 100.0 | 100.0 | 100.0 |
| KS93WGRC28 | *Pm20* | *V20* | 90.9 | 86.4 | 79.2 | 85.5 |
| Yangmai 5/sub.6v | *Pm21* | *V21* | 0.0 | 0.0 | 0.0 | 0.0 |
| Virest | *Pm22 (Pm1e)* | *V22 (V1e)* | 41.9 | 34.8 | 37.5 | 38.1 |
| Line81-7241 | *Pm23* | *V23* | 63.0 | 56.2 | 95.8 | 71.7 |
| 5P27 | *Pm30* | *V30* | 81.5 | 75.9 | 100.0 | 85.8 |
| Kenguia | *Pm4 + 8* | *V4 + 8* | 81.5 | 96.6 | 91.7 | 89.9 |
| Coker983 | *Pm5 + 6* | *V5 + 6* | 44.4 | 51.7 | 91.7 | 62.6 |
| Mission | *Pm4b + mli* | *V4b + mli* | 40.7 | 86.2 | 87.5 | 71.5 |
| Xiaobaidongmai | *PmXBD* | *VXBD* | 30.7 | 24.2 | 87.5 | 47.5 |
| Baimian 3 | *Pm4 + 2X* | *V4 + 2X* | 77.8 | 93.1 | 91.7 | 87.5 |
| CI12632 | *Pm2 + 6* | *V2 + 6* | 40.7 | 58.6 | 95.8 | 65.0 |
| Era | *Era* | *Era* | 37.0 | 86.2 | 87.5 | 70.2 |
| Funo | – | – | 100.0 | 100.0 | 100.0 | 100.0 |

## Virulence diversity and genetic diversity of *Bgt*

The virulence diversity "0, 1" matrix of these isolates was constructed based on the infection type of 80 *Bgt* isolates on 34 identified hosts. The phylogenetic tree clustered according to the similarity of infecting hosts is shown in Fig. 5. When the genetic similarity coefficient is 0.31, except for isolates L30, L31, L35, L42, and L45 from Liaoning; isolates H19, H21, and H29 from Heilongjiang, and C2-4 from Sichuan, 71 out of 80

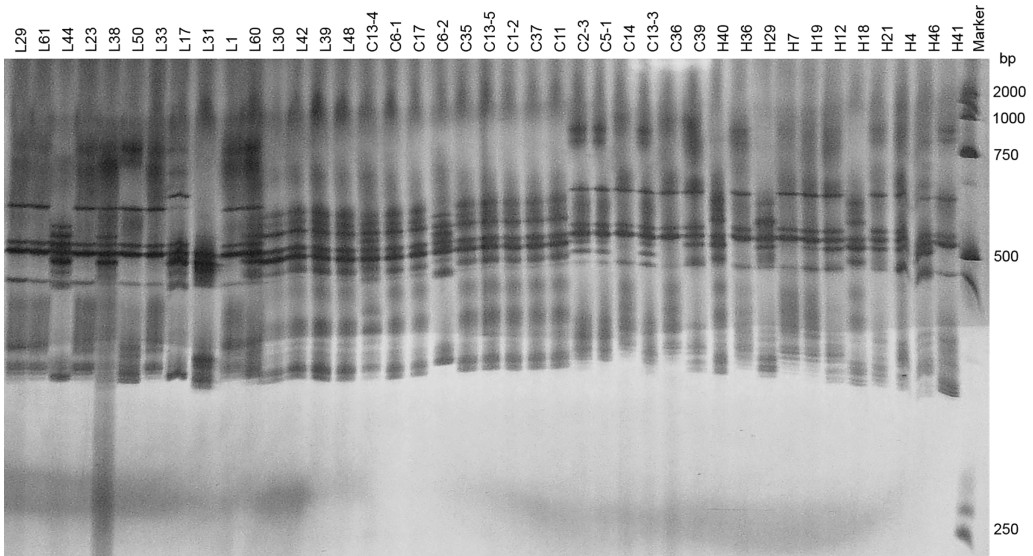

**Figure 1** Representative PAGE image of PCR products amplified by marker Blu SSR3–1-Blu SSR3–2 using *B. graminis* f. sp. *tritici* DNA.

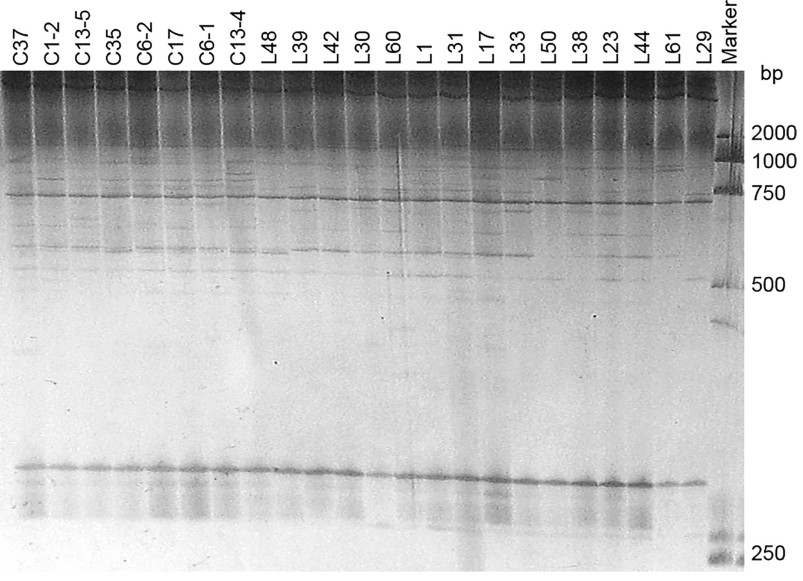

**Figure 2** Representative PAGE image of PCR products amplified by marker Blu SSR32–1-Blu SSR32–2 using *B. graminis* f. sp. *tritici* DNA.

isolates were divided into the same group. The PCA model described 34.0% variance. A total of 23.8% variance was described by Dim 1, and 11.2% was by Dim 2. All isolates were divided into three groups (Fig. 6).

The EST-SSR polymorphism and virulence diversity dendrograms of these isolated genes revealed that when the coefficient was <0.21, there were 10 clusters of genetic and virulence diversity. The positions of other isolates in the genetic polymorphism and virulence diversity dendrograms were different and did not correspond to each other.
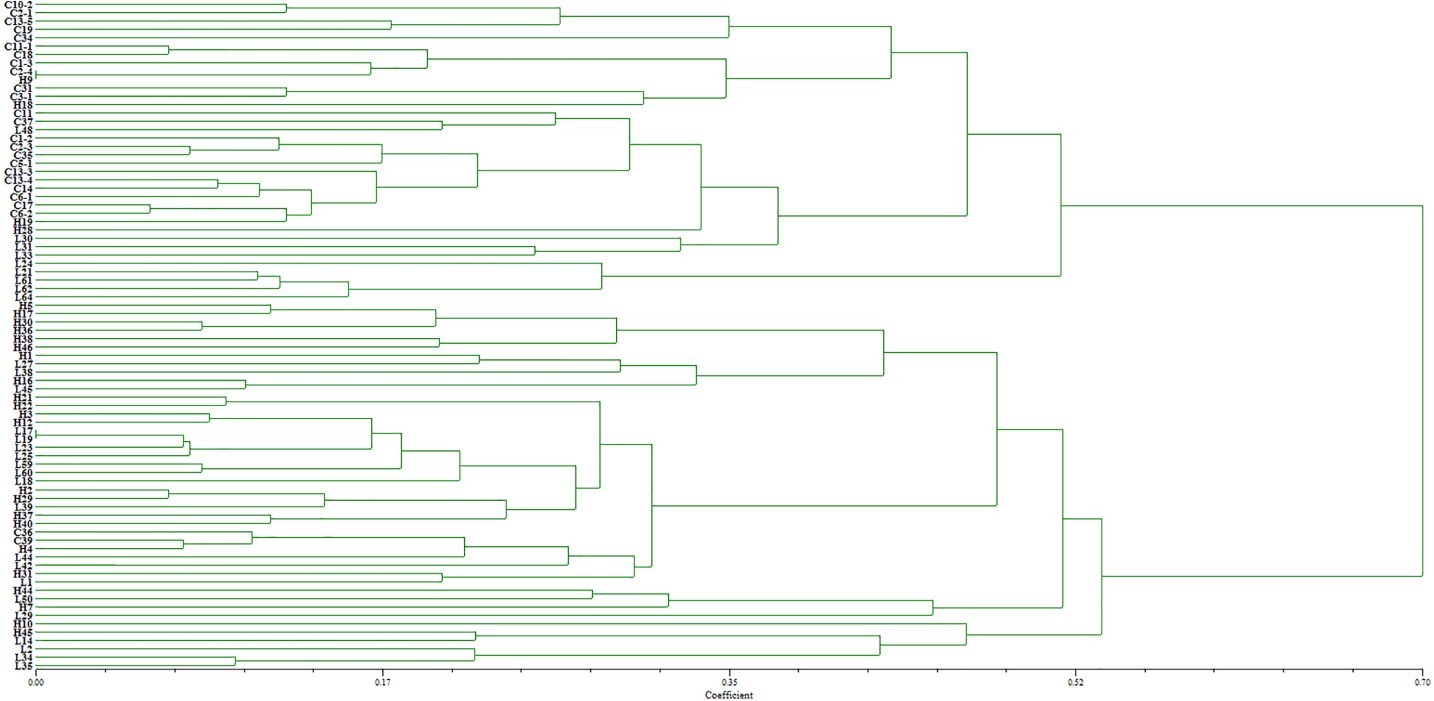

**Figure 3 Dendrogram of clustering analysis based on EST-SSR data for the genetic diversity of *B. graminis* f. sp. *tritici* isolates from different origins.**

Thus, there was a correlation between genetic polymorphism and virulence diversity of 36 isolates (45%) (Table S2). However, the EST-SSR polymorphism of isolated genes did not correspond to the virulence diversity of isolates in the single-gene lineage identification of hosts.

## DISCUSSION

Due to the gene-for-gene relationship between wheat and its fungal pathogen *Bgt*, studies of physiological race play an impressive role in monitoring the population dynamics of fungus. Continuous studies show that the virulence of *Bgt* is increasing year by year and cultivars carrying resistance genes tend to lose their effectiveness. In 2008–2009 the resistance genes *Pm4b*, *Pm2 + 6*, *Pm4 + 8*, *Pm12*, *Pm16*, *Pm18* (*Pm1c*), *Pm20*, *Pm21*, *Pm22* (*Pm1e*), and *Pm23* were effective against isolates from Northeast China (*Chi, 2009*). In 2011–2012, the resistance genes *Pm2*, *Pm4a*, *Pm4b*, *Pm12*, *Pm13*, *Pm16*, *Pm18* (*Pm1c*), *Pm19*, *Pm20*, *Pm21*, *Pm22* (*Pm1e*), *Pm23*, *and Pm5 + 6* were effective against isolates from Northeast China (*Chen et al., 2013*). In 2013–2014 the resistance genes *Pm13*, *Pm16*, *Pm18* (*Pm1c*), *Pm21*, *Pm22* (*Pm1e*), and *PmXBD* were effective against isolates from Northeast China. In the present study, we found that only resistance genes *Pm21* and *PmXBD* are effective against *Bgt* isolates collected in 2015 from Northeast China. Therefore, continuous virulence monitoring of *Bgt* can provide a reliable basis for breeding for disease resistance in Northeast China. Although the incidence rates of the virulence genes *V18 (V1c)* and *V22 (V1e)* increased to between 40% and 50%, their corresponding resistance genes still have a moderate value. Additionally, only *Pm21* constitutes an

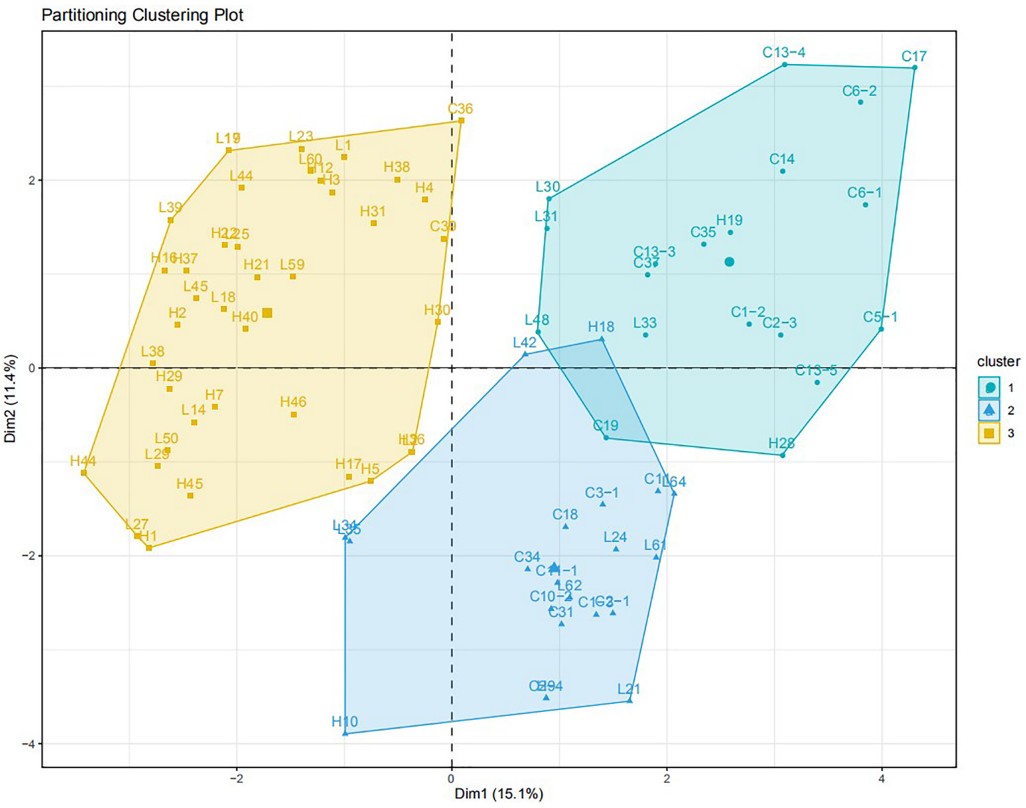

**Figure 4 PCA analysis for the genetic diversity of 80 *B. graminis* f. sp. *tritici* isolates.**

effective resistance gene towards isolates from Sichuan. The resistance genes *Pm18 (Pm1c)* and *Pm22 (Pm1e)* are of average resistance but can still be used. Gene *Pm21* is transferred from *Haynaldia villosa*, it has been widely used in a wheat breeding program in China (*Cao et al., 2011*; *Wu et al., 2019*). According to statistics, more than 10 commercial wheat have been released since 2002, with a planting area of more than $3.4 \times 10^6$ hm² (*Cao et al., 2011*). This gene encodes a typical CC-NBS-LRR (NLR for short) protein which recognizes the presence of specific pathogen 'avirulence' molecules and thus induces host defenses (*He et al., 2018*). As NLRs are recognition proteins, the resistance they control is almost always readily overcome by mutations in the pathogen's avirulence protein which prevent it from being recognized, so host defenses are not induced. For example, a few researchers reported isolates with virulence to the gene, which should be paid great attention (*Li et al., 2016*).

According to the occurrence of *Bgt* virulent isolates among different provinces, there is a regional difference between the distribution of virulence genes and the incidence rate. Results show that the incidence rates of *V13* in Heilongjiang and Liaoning were 18.5% and 62.1%, respectively. The incidence rate of *V2 + mld* in Heilongjiang and Liaoning was 22.2% and 79.3%, respectively, the incidence rate of *V2* in Heilongjiang and Liaoning in 2015 was 33.3% and 93.1%, and the incidence rate of *V8* in Heilongjiang and Liaoning was 33.3% and 86.2%, respectively. This indicates that *Pm2*, *Pm8*, *Pm13*, and *Pm2 + mld*
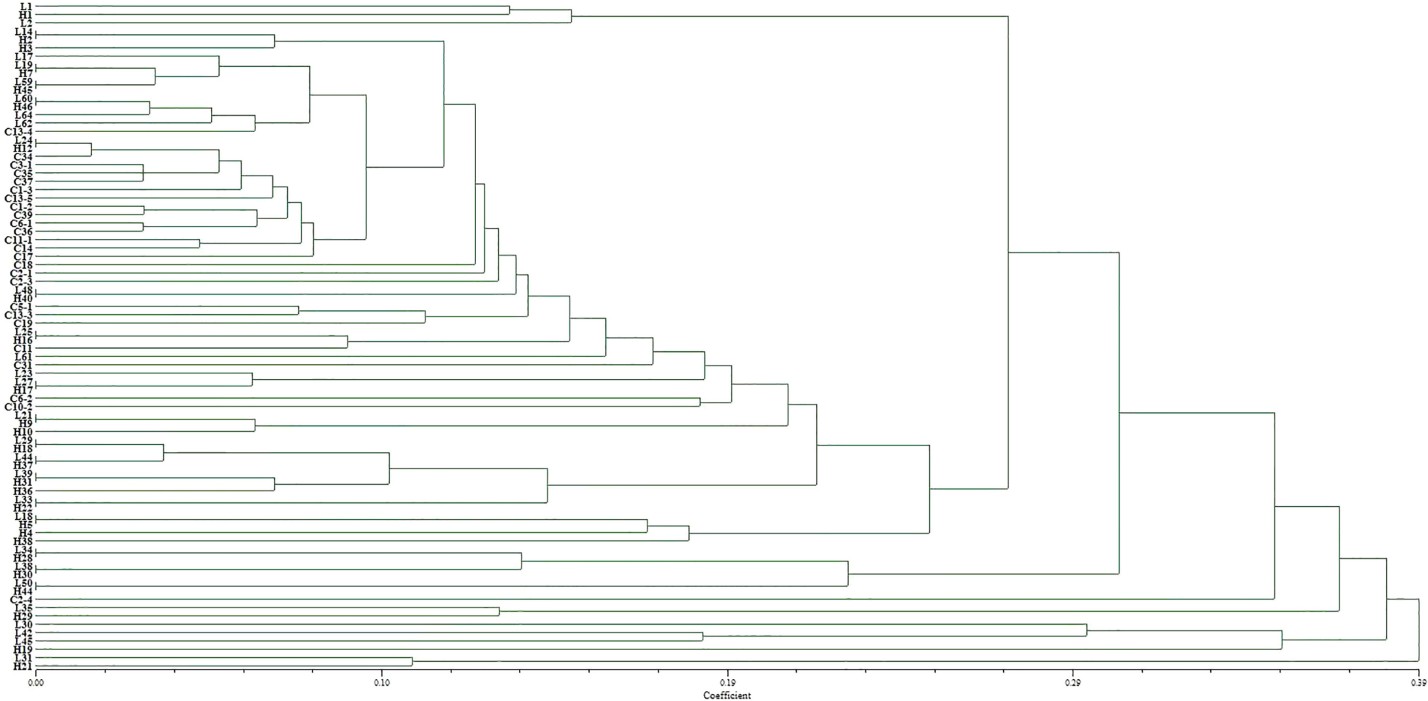

**Figure 5 Dendrogram of clustering analysis for the virulence diversity of 80 *B. graminis* f. sp. *tritici* isolates.**

are effective resistance genes in Heilongjiang but are ineffective when used alone in Liaoning. Similarly, the incidence rate of the virulence genes corresponding to the resistance genes *Pm4b* + *mli* and *Era* is much higher in Liaoning than in Heilongjiang. These resistance genes have almost completely lost their effectiveness in Liaoning, but still have some effectiveness in Heilongjiang. They still show clear superiority in the disease resistance breeding process of isolates from Heilongjiang.

Genetic diversity, which is also known as genetic polymorphism, is of great significance to species adapting to environmental changes and for survival and replication. With the development of molecular biology, molecular marker technology has been widely used in the genetic diversity analysis of plant pathogenic bacteria. Molecular marker technology was used to study the genetic diversity of *Bgt*, to explore its genetic variation and regional transmission, providing a prerequisite basis for the effective prevention and control of wheat powdery mildew and the rational distribution of disease-resistant cultivars (*Jia et al., 2007*; *Wang et al., 2001*; *Wolfe & Schwarzbach, 1978*). In this experiment, the genetic diversity of *Bgt* isolates from Northeast China and Sichuan was explored from the perspective of gene expression sequences using EST-SSR molecular marker technology. Genetic differences in the expressed sequences of *Bgt* from different regions in 2015 were analyzed. Virulence diversity and genetic diversity were also compared. The results showed that EST-SSR molecular marker technology appropriately revealed the genetic diversity of *Bgt*.

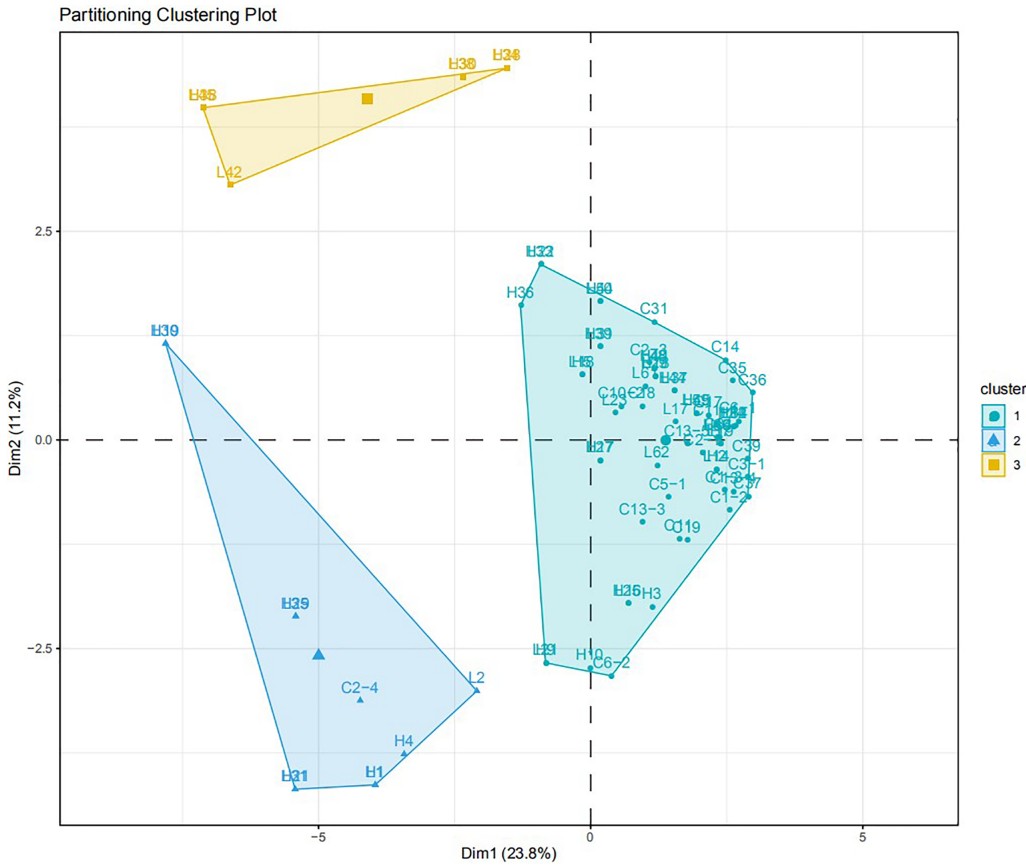

**Figure 6 PCA analysis for the virulence diversity of 80 *B. graminis* f. sp. *tritici* isolates.**

Based on the similarity coefficient, cluster analysis of *Bgt* isolates revealed that some isolates collected from Liaoning, Heilongjiang, and Sichuan in 2015 were clustered together, indicating that these isolates might have a certain degree of transmission and exchange, with more exchange rate for Liaoning and Heilongjiang isolates. However, at higher a genetic similarity coefficient isolates from different regions were clustered into small groups, indicating larger genetic differences among them.

When comparing the genetic diversity and host virulence polymorphism of *Bgt* with the virulence polymorphism on the host, we found that 18 out of 36 sample isolates were clustered together in genetic diversity and virulence diversity clusters in four different combinations. To some extent, the EST-SSR molecular marker technology revealed a correlation between *Bgt* genetic diversity and virulence diversity. However, the genetic diversity and virulence diversity of 55% of the isolates were different and did not correspond to each other.

## ACKNOWLEDGEMENTS
We are very grateful to Dr. Jincheng Zhou from Shenyang Agricultural University for his PCA analysis of the data.

### Funding

This study was supported by the Natural Science Foundation of Education Department of Liaoning Province (No. LJKZ0648) and the Natural Science Foundation of Liaoning Province (2020-MS-204). The funders had no role in study design, data collection and analysis, decision to publish, or preparation of the manuscript.

### Grant Disclosures

The following grant information was disclosed by the authors:
Natural Science Foundation of Education Department of Liaoning Province: LJKZ0648.
Natural Science Foundation of Liaoning Province: 2020-MS-204.

### Competing Interests

The authors declare that they have no competing interests.

### Author Contributions

- Yazhao Zhang analyzed the data, prepared figures and/or tables, and approved the final draft.
- Xianxin Wu performed the experiments, prepared figures and/or tables, and approved the final draft.
- Wanlin Wang performed the experiments, prepared figures and/or tables, and approved the final draft.
- Yiwei Xu analyzed the data, authored or reviewed drafts of the article, and approved the final draft.
- Huiyan Sun analyzed the data, prepared figures and/or tables, and approved the final draft.
- Yuanyin Cao conceived and designed the experiments, authored or reviewed drafts of the article, and approved the final draft.
- Tianya Li conceived and designed the experiments, prepared figures and/or tables, and approved the final draft.
- Mansoor Karimi-Jashni analyzed the data, authored or reviewed drafts of the article, and approved the final draft.

### Data Availability

The original, full-length gel of Figures 1 and 2 are available in the Supplemental File.

### Supplemental Information

Supplemental information for this article can be found online at http://dx.doi.org/10.7717/peerj.14118#supplemental-information.

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
