# Peer review of "Virulence characteristics of Blumeria graminis f. sp. tritici and its genetic diversity by EST-SSR analyses"

_PeerJ, doi:10.7717/peerj.14118_

## Round 0.1 · original submission · Major Revisions

Dear authors:

Thank you for submitting your manuscript to PeerJ for publication. Your manuscript had reviewed by two experts in your research area. Based on reviewers' recommendations and my assessment, your manuscript needs MAJOR revisions for publication.

The comments from reviewers are included at the bottom of this letter and in the attachments. Both reviewers comment on the important and good work but point out that the manuscript needs significant improvement. In particular, data analysis using robust cluster analysis software such as STRUCTURE or PCA is strongly recommended. Also, what were the novelty of your work and interpretation, biological significance, and accuracy of your research results? Major flaws of this paper include the writing. The manuscript can be improved by editing the English language throughout the manuscript. Please use the attached pdf file for further English editing.

Best regards,

Tika Adhikari

Reviewer 1 ·

Basic reporting

Powdery mildew caused by Blumeria graminis f. sp. tritici (Bgt) is one of the most important foliar diseases in wheat. Understanding the genetic diversity of Bgt population and their virulence to powdery mildew resistance genes are valuable for wheat breeders to select effective powdery mildew resistance genes (Pm) in the breeding program.

Experimental design

In the manuscript by Zhang et al., the authors tested the reaction patterns of 36 Pm gene differential lines to 80 Bgt isolates collected from Northeastern China (Liaoning and Heilongjiang provinces) and Sichuan province. The results provide some interesting information for the Bgt virulence of the three provinces to some of the Pm genes.

Validity of the findings

The genetic diversity of Bgt populations and their virulence to some of the known Pm genes were reported. However, some major concerns need to be addressed:
1. Table 2 listed the resistance reaction of the differential lines to the Bgt isolates. The authors described them as single gene lines (L129). However, some of the wheat lines contained more than two known Pm genes, such as, Pm4 +8, Pm5+6, etc. This description needs to be revised.
2. Some of the results in Table 2 are unreliable and need to be rechecked. For example, Normandie (Pm1+2+9) has distinct reaction pattern to Ulka/8cc and Maris Dove (the authors also wrote a wrong name of this wheat line!) that also contain Pm2 gene. Two lines contain Pm6, Timgalen (the authors also wrote a wrong name of this wheat line!) and Coker474, also showed different reactions to the Bgt isolates tested. Brigand (Pm16) and Chiyacao (Pm24) were proved to be effective Pm genes against almost all of the Bgt isolates collected from many regions of China (Ref: many publications). However, the authors identified a high percentage of virulent Bgt isolates to the two differential lines. Since functional markers for Pm2 and Pm24, and linked markers for Pm6 and Pm16 are available, the authors need to confirm the presence of the Pm genes in those wheat lines.
3. The authors presented three diversity trees (Figure 3, 4, 5). Figure 3 was based on the genotyping results of two EST-SSRs on 80 Bgt isolates. Why the authors not present a virulence tree of the 80 Bgt isolates? Instead, they selected 36 Bgt isolates to reconstruct a genetic diversity tree and a virulence tree?
4. In L73, XXW, WLW and TYL represent what?
5. The quality of the Figures is very low.

Reviewer 2 ·

Basic reporting

The language of this article is upto marked.

Experimental design

The authors used only NTSYSpc 2.10 statistical software for cluster analysis. The authors also need to use STRUCTURE software and PCA analysis for getting better grouping pattern. From these three analysis, the Authors will get effective link between the morphological data and molecular analysis.

Validity of the findings

• This article is not appropriate for the general reader. How reader understand the isolates of C17, C6-2 or L23, L60 ? So, need to be given all isolates information in Supplementary file.

Additional comments

This manuscript is not appropriate for publication in the current format. I am not convinced for publication in this high impact factor journal. Need major revisions. Some instructions are given below:

• Similar works have been done by Wu et al 2018. Virulence structure and its genetic diversity analyses of Blumeria graminis f. sp. tritici isolates in China. I didn’t get any new presentation style from that paper. They have used 2 primers pair out of 7. In this article, the authors used 2 primers pairs out of the same 7 primers pairs. The molecular analysis also same with Wu et al paper. Need to clarify: Why the authors didn’t use all of the 7 primers? How many polymorphic alleles have been got in each primer pairs, need to be mentioned in manuscript?
• The Authors wrote that the primer pairs Blu SSR3-1-Blu SSR3-2 and Blu SSR29-1-Blu SSR29-2 were chosen but after that showed Fig 1 and Fig 2 the primer Blu SSR3-1, Blu SSR3-2 and Blu SSR32-1- plus Blu SSR32-2. Need to clarify?
• In Materials and Methods section the Author’s wrote two collection area Liaoning, Heilongjiang. Need to clarify?
• How many isolates were used for each location need to be written? The isolates information should be given as a supplementary file. The information should be included; collection location, collection date, longitude, latitude, elevation etc
• The authors can make clustering with virulence diversity using NTSYSpc 2.10 for all 80 isolates, no need sampling from each location. In that case Fig 4 can be excluded. It will be better for comparison between virulence analysis and genetic analysis.
• This article is not appropriate for the general reader. How reader understand the isolates of C17, C6-2 or L23, L60 ? So, need to be given all isolates information in Supplementary file.
• The authors used only NTSYSpc 2.10 statistical software for cluster analysis. The authors also need to use STRUCTURE software and PCA analysis for getting better grouping pattern. From these three analysis, the Authors will get effective link between the morphological data and molecular analysis.
• The authors can make clustering with virulence diversity using NTSYSpc 2.10 for all 80 isolates, no need sampling from each location. In that case Fig 4 can be excluded. It will be better for comparison between virulence analysis and genetic analysis.

Annotated reviews are not available for download in order to protect the identity of reviewers who chose to remain anonymous.

---

## Round 0.2 · Minor Revisions

Dear Dr. Li:

Thank you for revising and submitting your manuscript entitled "Virulence characteristics of Blumeria graminis f. sp. tritici and its genetic diversity by EST-SSR analyses". It further requires Minor Revisions.

I would appreciate it if you could revise it and submit it at your earliest convenience. Thank you.

Best regards,

Tika Adhikari

Reviewer 1 ·

Basic reporting

The authors improved the English writing. The contents of the manscript are improved.

Experimental design

My major concerns were answered. Some uncertain data was deleted and not used in the revised version.

Validity of the findings

The information is valuable for the society

Additional comments

The Pm2 functional marker was recommand to recheck the availability of Pm2 in the tested Normandie, Ulka/8cc, and Maris Dove.

Reviewer 2 ·

Basic reporting

The language of this article is upto marked.

Experimental design

Experimental design is sufficient

Validity of the findings

It is sufficient

Additional comments

I have given comments on Manuscript and Table 1 file. Please see the attachments.

Annotated reviews are not available for download in order to protect the identity of reviewers who chose to remain anonymous.

---

## Round 0.3 · accepted · Accept

Dear Dr. Li:

Thank you for your submission to PeerJ.

I am writing to inform you that your manuscript - Virulence characteristics of Blumeria graminis f. sp. tritici and its genetic diversity by EST-SSR analyses - has been Accepted for publication. Congratulations!

Best regards,

Tika Adhikari